# The Polymorphisms of the Peroxisome-Proliferator Activated Receptors’ Alfa Gene Modify the Aerobic Training Induced Changes of Cholesterol and Glucose

**DOI:** 10.3390/jcm8071043

**Published:** 2019-07-17

**Authors:** Agnieszka Maciejewska-Skrendo, Maciej Buryta, Wojciech Czarny, Pawel Król, Michal Spieszny, Petr Stastny, Miroslav Petr, Krzysztof Safranow, Marek Sawczuk

**Affiliations:** 1Department of Molecular Biology, Faculty of Physical Education, Gdansk University of Physical Education and Sport, 80-336 Gdansk, Poland; 2Department of Anatomy and Anthropology, Faculty of Physical Education, University of Rzeszow, 35-310 Rzeszow, Poland; 3Institute of Sports, Faculty of Physical Education, University of Physical Education and Sport, 31-571 Kraków, Poland; 4Department of Sport Games, Faulty of Physical Education and Sport, Charles University, 162-52 Prague, Czech Republic; 5Department of Biochemistry and Medical Chemistry, Pomeranian Medical University, 70-204 Szczecin, Poland; 6Unit of Physical Medicine, Faculty of Tourism and Recreation, Gdansk University of Physical Education and Sport, 80-336 Gdansk, Poland

**Keywords:** human performance, aerobic training, genetic predisposition, lipid metabolism, glucose tolerance, VO_2_max, mitochondria activity, cholesterol levels

## Abstract

Background: PPARα is a transcriptional factor that controls the expression of genes involved in fatty acid metabolism, including fatty acid transport, uptake by the cells, intracellular binding, and activation, as well as catabolism (particularly mitochondrial fatty acid oxidation) or storage. *PPARA* gene polymorphisms may be crucial for maintaining lipid homeostasis and in this way, being responsible for developing specific training-induced physiological reactions. Therefore, we have decided to check if post-training changes of body mass measurements as well as chosen biochemical parameters are modulation by the *PPARA* genotypes. Methods: We have examined the genotype and alleles’ frequencies (described in *PPARA* rs1800206 and rs4253778 polymorphic sites) in 168 female participants engaged in a 12-week training program. Body composition and biochemical parameters were measured before and after the completion of a whole training program. Results: Statistical analyses revealed that *PPARA* intron 7 rs4253778 CC genotype modulate training response by increasing low-density lipoproteins (LDL) and glucose concentration, while *PPARA* Leu162Val rs1800206 CG genotype polymorphism interacts in a decrease in high-density lipoproteins (HDL) concentration. Conclusions: Carriers of *PPARA* intron 7 rs4253778 CC genotype and Leu162Val rs1800206 CG genotype might have potential negative training-induced cholesterol and glucose changes after aerobic exercise.

## 1. Introduction

PPARα is a transcriptional factor that controls the expression of genes involved in fatty acid metabolism, including fatty acid transport, uptake by the cells, intracellular binding, and activation, as well as catabolism (particularly mitochondrial fatty acid oxidation) or storage [1]. PPARα is expressed at moderate levels, mainly in the liver and skeletal muscles, but also in the heart, kidney, brown fat, and large intestine [2,3,4]. PPARα-dependent transcriptional activity results from a direct interaction of the nuclear receptor with its ligands [5]. The primary natural PPARα ligands are unsaturated fatty acids that directly bind to the PPARα via ligand binding domain (LBD) and enable its heterodimerization with the retinoid X receptor (RXR)-α. Such PPAR:RXR complex binds via PPARα DNA binding domain (DBD) to the PPRE (peroxisome proliferator response element) sequence in the promoter region of target genes [6]. In comparison with the unsaturated fatty acids, saturated fatty acids are poor PPAR ligands [7]. Moreover, synthetic compounds, such as hypolipidemic agents, prostaglandin 12 analogs, leukotriene B4 analogs, leukotriene D4 antagonist, carnitine palmitoyl transferase I (CPT1) inhibitors, fatty acyl-CoA dehydrogenase inhibitors, can activate PPAR [8,9,10]. It is worth noting that an alternative activation pathway of PPAR:RXR may also occur through ligand binding to RXR [11,12]. In addition to ligand-dependent activation, PPARα may also be regulated by insulin-induced trans-activation that occurs through the phosphorylation of two mitogen-activated protein (MAP) kinase sites at positions 12 and 21 located in the activation function (AF)-1-like domain within PPARα receptor [13].

The *PPARA* gene has been mapped on the human chromosome 22 (locus 22q12-q13.1) and comprises a total of eight exons encoding PPARα protein [14]. Within the entire gene, several polymorphic sites have been identified, with the most studied variant, a missense mutation C/G (rs1800206), resulting in Leu162Val amino acids substitution. This polymorphic site is located in the exon 5 of the *PPARA* gene that encodes the second zinc finger of the DNA binding domain in the PPARα protein. Despite the fact that Leu to Val is a conservative change, this amino acids substitution has functional consequences on protein activity, because the 162 position is next to a cysteine which coordinates the zinc atom and, at the same time, Leu162Val is located upstream of a region determining the specificity and polarity of PPARα binding to different PPREs [15]. In vitro experiments revealed that PPARα isoform with Val in the 162 position has increased PPRE-dependent transcriptional activity compared with the PPARα isoform with Leu in the same position when treated with the PPARα ligand [16]. Interestingly, observed differences were ligand concentration-dependence: At higher concentrations of the ligand, the 162Val variant’s transactivation activity was five-fold greater as compared with Leu162 variant [17]. In addition, in vivo observations confirmed that Leu162Val polymorphism exerts an effect on plasma lipoprotein–lipid profile. Carriers of the minor G allele (for Val in the 162 position) compared with homozygotes of the C allele (for Leu162) had significantly higher concentrations of plasma total and low-density lipoproteins (LDL)-apolipoprotein B as well as and LDL cholesterol [18]. In Type II diabetic patients, G allele carriers had higher levels of total cholesterol, high-density lipoproteins (HDL) cholesterol, and apoAI [16]. Moreover, G allele carriers were characterized by a better response to lipid-lowering drugs, showing a greater lowering effect with regard to total cholesterol and non-HDL-cholesterol than C allele homozygotes treated with the same drug [19]. Furthermore, other studies revealed that Leu162Val polymorphism influences the conversion from impaired glucose tolerance to type 2 diabetes [20] as well as being associated with progression of coronary atherosclerosis and the risk of coronary artery disease [21].

The second polymorphic site that has been studied in many contexts is a C/G substitution in *PPARA* intron 7 (rs4253778). It was described for the first time in 2002 in the publications focused on genetic modulators influencing the progression of coronary atherosclerosis and the risk of coronary artery disease [21] as well as left ventricular growth [22]. It has been revealed that *PPARA* rs4253778 polymorphism influences human left ventricular growth observed in response to exercise and hypertension: The C allele carriers had significantly higher left ventricular mass. Moreover, the observed effect was additive: CC homozygotes had a 3-fold greater, and GC heterozygotes had a 2-fold greater increase in left ventricular mass than G allele homozygotes [22]. Taking into account that one of the molecular adaptations described in the hypertrophied heart is reduced PPARα activity [23] and, at the same time, an increase in glucose utilization and a decrease in fatty acid oxidation (FAO) is observed [24,25], it has been hypothesized that the *PPARA* intron 7 C allele affects PPARα function and is connected with downregulation of the expression of mitochondrial FAO enzymes, leading to reduced FAO and impaired cellular lipid homeostasis [22]. Study with diabetic patients has confirmed that C allele carriers are characterized by reduced the lipid-lowering response to fenofibrate treatment in comparison with GG homozygotes [26]. Next, studies with athletes representing different sports disciplines revealed that GG homozygotes were more prevalent in the groups of endurance-type athletes engaged in prolonged aerobic exertion [27,28], while the C allele was frequently observed in power-oriented athletes who were involved in shorter and very intense anaerobic exertion [29]. These results were partly explained by muscle biopsies showing the association between *PPARA* rs4253778 polymorphism and fiber type composition, particularly the correlation between G allele and increased proportion of type I (oxidative) fibers as well as the association of the C allele with the propensity to skeletal muscle hypertrophy, and a facilitation of glucose utilization in response to anaerobic exercise [29].

All the aforementioned facts suggest that *PPARA* polymorphisms may be crucial for maintaining lipid homeostasis and in this way, be responsible for developing specific training-induced physiological reactions. Therefore, we have decided to check if post-training changes of body composition measurements, as well as chosen biochemical parameters (LDL, HDL, glucose), are modulated by the *PPARA* genotypes. To test this hypothesis, we have examined the genotype and alleles’ frequencies (described in *PPARA* rs1800206 and rs4253778 polymorphic sites) in female participants engaged in a 12-week training program.

## 2. Experimental Section

### 2.1. Ethics Statement

The procedures followed in the study were conducted ethically according to the principles of the World Medical Association Declaration of Helsinki and ethical standards in sport and exercise science research. The study was approved by the Ethics Committee of the Regional Medical Chamber in Szczecin (Approval number 09/KB/IV/2011). All participants were given a consent form and a written information sheet concerning the study, providing all pertinent information (purpose, procedures, risks, and benefits of participation). The experimental procedures were conducted in accordance with the set of guiding principles for reporting the results of genetic association studies defined by the Strengthening the Reporting of Genetic Association studies (STREGA) Statement [30].

### 2.2. Participants

Out of 201 recruited Polish Caucasian women (range 19–24 years) we have obtained 182 full sets of pre-training and post-training body composition and biochemical data in those who completed a 12-week training program. From these 182 participants, the genetic material was isolated and 168 samples (age 21.6 ± 1.3 years, body mass 60.6 ± 7.6 kg, 21.6 ± 2.4) were successfully genotyped for PPARA rs1800206 and rs4253778. None of the included individuals had engaged in regular physical activity in the previous 6 months. The level of physical activity over the last 6 months has been estimated in every participant according to Global Physical Activity Questionnaire (GPAQ) as well as the individual recording of the subject’s own activity, such as direct observation and activity diaries [31]. They had no history of any metabolic or cardiovascular diseases. Participants were nonsmokers and refrained from taking any medications or supplements known to affect metabolism. Before the training phase, all participants were included in a dietary program and had received an individual dietary plan. For every participant, the Basal Metabolic Rate (BMR) as well as the Physical Activity Level (PAL, calculated as the ratio of Total Energy Expenditure (TEE) to BMR), was defined. Every participant was asked to keep a balanced diet customized for the individual’s PAL coefficient and body mass according to nutrition standards described for the Polish population [32] during the study and for 2 months before the study. The participants were asked to keep a food diary every day. Weekly consultations were held in which the quality and quantity of meals were analyzed and, if necessary, minor adjustments were made. The nutrition and general lifestyle conditions for all participants during the training phase were considered as similar. During the last weekly session before the 12-week training program, the participants underwent the graded exercise VO_2_max test and body composition screen.

### 2.3. Training Intervention

Maximum heart rate (HRmax) was calculated directly in every subject by a continuous graded exercise test on an electronically braked cycle ergometer (Oxycon Pro, Erich JAEGER GmbH, Hoechberg, Germany) which was performed to determine their aerobic capacity (VO2max). The heart rate (HR) at each step of the training program was measured in every subject using HR personal monitoring devices (Polar T31 straps and CE0537 Watches, Lake Success, NY, USA) with customized setup. The training stage was preceded by a week-long familiarization stage, when the examined women exercised 3 times a week for 30 min, at an intensity of about 50% of their HRR (HR Reserve) calculated according to the Karvonen formula. After the week-long familiarization stage, proper training has started. Each training unit consisted of a warm-up routine (10 min), the main aerobic routine (43 min), and stretching and breathing exercise (7 min). The main aerobic routine was a combination of two alternating styles—low and high impact as described by Zarebska et al. [33,34,35]. Low impact style comprised movements with at least one foot on the floor at all times, whereas high impact styles included running, hopping, and jumping with a variety of flight phases [36]. Music of variable rhythm intensity (tempo) was incorporated into both styles. A 12-week program of low–high impact aerobics was divided as follows: (1) 3 weeks (9 training units), 60 min each, at about 50–60% of HRR, music tempo 135–140 BPM (beats per min), (2) 3 weeks (9 training units), 60 min each, at 55–65% of HRR, music tempo 135–140 BPM, (3) 3 weeks (9 training units), 60 min with the intensity of 60–70% of HRR, music tempo 140–152 BPM, and (4) 3 weeks (9 training units), 60 min with an intensity of 65–75% of HRR, music tempo 140–152 BPM. All 36 training units were administered and supervised by the same instructor.

### 2.4. Body Composition Measurements

Body mass and body composition were assessed by the bioimpedance method (body’s inherent resistance to an electrical current) with the use of the electronic scale “Tanita TBF 300M” (Horton Health Initiatives, Orland Park, IL, USA) as described by Zarebska et al. [33]. The device was plugged in and calibrated with the consideration of the weight of the clothes (0.2 kg). Afterward, data regarding age, body height, and sex of the subject were inserted. Then, the subjects stood on the scale with their bare feet on the marked places without leaning any body part. The device analyses body composition based on the differences in the ability to conduct electrical current by body tissues (different resistance) due to different water content. Body mass and body composition measurements were taken with the use of the electronic scale “Tanita” are as follows: total body mass (kg), fat free mass (FFM, kg), fat mass (kg), body mass index (BMI = body mass (kg)/(body height (m))^2^, in kg.m^−2^), tissue impedance (Ohm), total body water (TBW, kg), and basal metabolic rate (BMR, kJ).

### 2.5. Biochemical Analyses

Fasting blood samples were obtained in the morning from the elbow vein before the start of the aerobic fitness training program and repeated at the 12th week of this training program (after the 36th training unit). Compete blood samples (taken before and after 12-week training period) were obtained for 182 participants. The analyses were performed immediately after the blood collection, as described by Leońska-Duniec et al. [37]. Blood samples from each participant were collected in 2 tubes. For biochemical analyses, a 4.9 mL·S-Monovette tube with ethylenediaminetetraacetic acid (K 3 EDTA; 1.6 mg EDTA/mL blood) and separating gel (SARSTEDT AG and Co., Nümbrecht, Germany) were used. Blood samples for biochemical analyses were centrifuged 300× g for 15 min at room temperature to receive blood plasma. All biochemical analyses were conducted using Random Access Automatic Biochemical Analyzer for Clinical Chemistry and Turbidimetry A15 (BIO-SYSTEMS S.A., Barcelona, Spain). Blood plasma was used to determine lipid profile: triglycerides (TGL), total cholesterol, high-density lipoproteins (HDL) and low-density lipoproteins (LDL) concentrations. Plasma TGL and total cholesterol concentrations were determined using a diagnostic colorimetric enzymatic method according to the manufacturer’s protocol (BioMaxima S.A., Lublin, Poland). The manufacturer’s declared intra-assay coefficients of variation (CV) of the method were <2.5% and <1.5% for the TGL and total cholesterol determinations, respectively. HDL plasma concentration was determined using the human anti-β-lipoprotein antibody and colorimetric enzymatic method according to the manufacturer’s protocol (BioMaxima S.A.). The manufacturer’s declared intra-assay CV of the method was <1.5%. Plasma concentrations of LDL were determined using a direct method according to the manufacturer’s protocol (PZ Cormay S.A., Lomianki, Poland). The manufacturer’s declared intra-assay CV of the method was 4.97%. All analysis procedures were verified with the use of a multi-parametric control serum (BIOLABO S.A.S, Maizy, France), as well as control serum of normal level (BioNormL) and high level (BioPathL) lipid profiles (BioMaxima S.A.).

### 2.6. Genetic Analyses

The buccal cells donated by the subjects were collected in Resuspension Solution (GenElute Mammalian Genomic DNA Miniprep Kit, Sigma-Aldrich Chemie Gmbh, Munich, Germany) with the use of sterile foam-tipped applicators (Puritan, Holbrook, NY 11741, USA). DNA was extracted from the buccal cells using a GenElute Mammalian Genomic DNA Miniprep Kit (Sigma-Aldrich Chemie Gmbh, Munich, Germany) according to the manufacturer’s protocol. DNA isolates were evaluated for quantity, quality, and integrity of DNA using the spectrophotometer BioPhotometer Plus (Eppendorf, Wesseling-Berzdorf, Germany). Only 168 isolates passed the evaluation and were used for subsequent genotyping.

To discriminate *PPARA* I7 rs4253778 (G > C) as well as Leu162Val rs1800206 (C > G) alleles, TaqMan Pre-Designed SNP Genotyping Assays were used (Applied Biosystems, Waltham, MA, USA) (assay IDs: C___2985251_10 and C___8817670_20, respectively) including primers and fluorescently labeled (FAM and VIC) MGBTM TaqMan probes to detect alleles. All samples were genotyped in duplicate on a StepOne Real-Time Polymerase Chain Reaction (RT-PCR) instrument (Applied Biosystems, Waltham, MA, USA) as previously described [37]. PCR products were then subjected to Endpoint-genotyping analysis using an allelic discrimination assay at StepOne Software v2.3 (Applied Biosystems, Carlsbad, CA, USA) to measure the relative amount of allele-specific fluorescence (FAM or VIC), which leads directly to the determination of individual genotypes. Genotypes were assigned using all of the data from the study simultaneously.

### 2.7. Statistical Analyses

Allele frequencies were determined by gene counting. An χ^2^ test was used to test the Hardy–Weinberg equilibrium. To examine the hypothesis that the *PPARA* I7 rs4253778 polymorphism modulate training response, we conducted a repeated measure 2 × 3 ANOVA for genes and 2 × 2 ANOVA for alleles comparison with one between-subject factor (*PPARA* I7 rs4253778 genotype: GG vs. GC vs. CC, GG vs. GC + CC, GG + GC vs. CC) and one within-subject factor (time: before training versus after training) for twelve dependent variables. To examine the hypothesis that *PPARA* Leu162Val rs1800206 modulate training response, we conducted a repeated measure of 2 × 2 ANOVA. Kolmogorov–Smirnov test was used to check for data normality, Mauchly’s test for data sphericity, and a post hoc Tukey test was applied when interaction was significant and was used to perform pair-wise comparisons. The effect size (partial eta squared–η^2^) of each test was calculated for all analyses and was classified according to Larson-Hall [38], where η^2^: 0.01, 0.06, 0.14 were estimated for small, moderate, and large effect, respectively. All statistics were performed in STATISTICA software (version 13; StatSoft, Tulsa, OK, USA) with the level of statistical significance set at *p* < 0.05.

## 3. Results

*PPARA* I7 rs4253778, as well as Leu162Val rs1800206 genotypes, conformed to Hardy–Weinberg equilibrium (*p* = 0.887 and *p* = 0.572, respectively) and phenotype outcomes were normally distributed, with no disruption of sphericity (Appendix A). The genotyping error was assessed as 1%, while the call rate (the proportion of samples in which the genotyping provided unambiguous reading) exceeded 95%.

The ANOVA showed genotype × training interactions in *PPARA* I7 rs4253778 for LDL (F_1, 165_ = 5.12, *p* = 0.025, η^2^ = 0.03) (Table 1), where post hoc test showed that LDL increased in *PPARA* I7 rs4253778 CC homozygotes after training intervention (79.17 ± 14.16 vs. 95.48 ± 15.35 mg/dL), which did not appear in other genotypes (Figure 1). ANOVA in *PPARA* I7 rs4253778 allele × training interactions showed differences in LDL (F_1, 166_ = 4.59, *p* =.034, η^2^ = 0.03), where post hoc analyses showed that CC homozygotes and not G allele carriers increased the LDL concentration after training intervention (Table 1).

Other *PPARA* I7 rs4253778 genotype × training interactions were found in glucose plasma concentrations (F_2, 165_ = 3.99, *p* = 0.02, η^2^ = 0.05) (Table 1), where post hoc showed that carriers of GG and GC genotypes decreased glucose concentration over the period of training (78.83 ± 10.20 vs. 76.44 ± 10.21 and 77.62 ± 8.92 vs. 73.34 ± 9.28, respectively, mg/dL), while for CC homozygotes were characterized by the opposite effect of training and demonstrated a significant increase of glucose concentration (70.50 ± 7.76 vs. 78.17 ± 12.58 mg/dL) (Figure 2) (Table 1). Furthermore, ANOVA in PPARA I7 rs4253778 allele × training interactions showed differences in glucose concentration (F_1, 166_ = 6.68, *p* = 0.011, η^2^ = 0.06), where post hoc showed that G allele carriers decreased glucose concentration and CC homozygotes increased glucose concentration (Table 1).

HDL plasma concentration resulted in statistical differences in *PPARA* Leu162Val rs1800206 genotype (F_1, 166_ = 22.68, *p* < 0.001, η^2^ = 0.12) and training interactions (F_1, 166_ = 6.30, *p* = 0.013, η^2^ = 0.04) (Table 2), where post hoc showed that HDL decreased in both *PPARA* Leu162Val rs1800206 genotypes (CC and CG) in the course of training (65.12 ± 13.51 vs. 61.78 ± 13.48 and 64.82 ± 11.83 vs. 54.03 ± 12.08, respectively) (Figure 3) and this training decrease was bigger in CG genotype (only G allele carriers) when compared to CC homozygotes (Table 2). There were no other effects of training or *PPARA* Leu162Val rs1800206 genotypes observed in dependent variables.

Our statistical analyses revealed that *PPARA* intron 7 rs4253778 polymorphism modulate training response in reference to plasma LDL and glucose concentration, while *PPARA* Leu162Val rs1800206 polymorphism interacts with HDL concentration. Taken together, our results supply additional information about the potential role played by genetic variants described in the *PPARA* gene in training-induced biochemical changes.

## 4. Discussion

This study aimed to check if post-training changes of body mass measurements, as well as chosen biochemical parameters observed in physically active women, are modulated by specific genotypes. The verified hypothesis assumed that in the presence of specific genotypes and alleles in the *PPARA* gene would influence the post-training response observed in biochemical parameter changes in the course of the 12-week training program. Taking onto account *PPARA* intron 7 (rs4253778) genotype × training interactions, there were two statistically significant effects: for LDL and glucose plasma concentrations. For all genotypes, a slight increase in LDL level was observed. However, the rise of LDL concentration reached the highest point in intron 7 CC homozygotes when compared to G allele carriers. Moreover, the CC homozygotes were characterized by an unexpected increase in glucose plasma concentration, while in GC and GG participants, the reverse trend of decreasing glucose concentration was noted. It is commonly expected that LDL, as well as glucose plasma levels, would decrease after regular physical activity [39]. However, more detailed analyses revealed that beneficial changes of the lipid profile are achieved only when the intensity of training is moderate (the exercises are performed below the anaerobic threshold), while the training above the anaerobic threshold intensity may not lead to such healthy effects; what is more it may even reverse these beneficial trends in the context of plasma lipid concentrations [40]. A meta-analysis of studies on the impact of aerobic training on plasma HDL concentration revealed that the minimum duration of aerobic exercises necessary for achieving the beneficial effect of HDL level elevation is about 120 minutes per week, which is an equivalent of 900 kcal energy expenditure [41]. In the exercise protocol used in our study, we had about 180 min exercises per week, and the intensity of the exercises was gradually increased from 50% to 60% to 65% to 75% heart rate reserve. Each training unit consisted of a warm-up, the main aerobic routine, and the ending phase, including stretching and a breathing exercise [33,34,35]. The structure of the main routine that was a combination of two alternating styles of low and high intensity may resemble interval training, in which the high-intensity workouts are similar to anaerobic exercises, while low-intensity sets correspond to a restitution phase. The summary volume of aerobic exercises probably was not enough to achieve the expected beneficial changes in the lipid profile and glucose level, especially in *PPARA* intron 7 CC homozygotes.

The functional role of *PPARA* intron 7 polymorphism was suggested for the first time in the prospective study of healthy middle-aged men in the United Kingdom [21] as well as in the study of male British Army recruits undergoing a 10-week physical training program [22]. It has been demonstrated that intron 7 C allele is associated with progression of atherosclerosis [21] and is positively correlated with left ventricular growth in response to exercise [22]. Based on the results of the studies showing that hypertrophied heart is characterized by reduced PPARα activity [23] and, in the same time, downregulation of the expression of mitochondrial FAO enzymes [25] with accompanying increases in expression of genes encoding glycolytic enzymes [24], it has been speculated that intron 7 C allele is responsible for lowering the expression of the *PPARA* gene and in this way, is indirectly connected with downregulation of the expression of key metabolic enzymes, leading to impairment of cellular lipid and glucose homeostasis. Another issue is, of course, the question of how the polymorphism located in the non-coding region influences the gene’s expression. One possible answer could be that *PPARA* intron 7 alleles are not direct casual variants, but are rather in linkage disequilibrium with an unidentified polymorphism (within the *PPARA* gene or in its regulatory region) that alters encoded protein levels and, as a consequence, may change the expression of PPARα target genes [27,29,42,43]. There is also a hypothesis that, considering this SNP location, the *PPARA* intron 7 polymorphism may change and disrupt a microRNA site [44].

Considering that the proper expression of the *PPARA* gene, necessary for maintaining the appropriate level of PPARα protein, is crucial for regulation of carbohydrate/lipid metabolism, and that the intron 7 C allele may affect this expression process, it is may be expected that the lipid profile and glucose levels would be altered in C allele carriers. Indeed, our results seem to confirm this assumption because we have observed that LDL and glucose levels in CC homozygotes were different from the normal range, with the surprising effect of a post-training increase of plasma glucose concentration as well as the highest rise of LDL levels observed in CC participants. These results suggest that *PPARA* intron 7 CC genotype may be in the group of disadvantageous factors responsible for developing unexpected post-training effects. Probably the CC homozygotes should undergo a different training program with increased volume of aerobic exercises to achieve the expected beneficial results.

When *PPARA* Leu162Val (rs1800206) genotype × training interactions were taken into account, only one statistically significant effect was observed: for post-training changes of HDL levels. In the case of both recorded measurements in these SNP genotypes (CC and CG) we have observed a slight decrease of HDL levels. However, this lowering effect was more pronounced in G allele carriers, in which at least half of the PPARα protein amount comprised the Val amino acid in the 162 position. It is worth noting that rs1800206 GG homozygotes are very rare in the human population, in our study, there were no such individuals in the whole study group.

The C→G substitution, described as *PPARA* rs1800206 polymorphism, is placed within the coding region of the gene, what makes it functional “by definition”, causing an amino acid change in the 162 position (Leu162Val) that is located within DNA binding domain (DBD) of the PPARα protein [18]. DBD is directly involved in the interaction between the PPARα transcription factor and PPRE sequences in the promoter region of target genes [6]. Detailed in vitro analyses revealed that PPARα constructs with Val amino acid in the 162 position is activated by the endogenous ligands to a lesser extent when compared with “wild type” PPARα with Leu amino acid residue in the same localization [45]. The PPARα Leu form, that is produced in CC homozygotes, is considered as an active form of this transcriptional factor, displaying a higher transcriptional activity [45] and being able to stimulate the expression of the genes encoding β-oxidation enzymes more efficiently, which cause the shift of the metabolic balance toward catabolic pathways [46,47]. 

In vivo studies have shown that Leu162Val polymorphism is associated with total plasma cholesterol [16,48,49], LDL [18,49], HDL [16], as well as apolipoprotein B (apoB) [18,48,49], apolipoprotein A-I (apoA-I) [16], and apolipoprotein C-III (apoC-III) [49] concentrations. Moreover, Leu162Val is involved in diabetes and arteriosclerosis progression [16,20,21]. To be more specific, the studies of diabetic patients and non-diabetic subjects revealed that the rs1800206 G allele (also designed as the 162Val allele) carriers were characterized by higher levels of plasma total cholesterol, LDL, and apoB levels in comparison with CC homozygotes [18]. The large population-based study confirmed that the presence of the rs1800206 G allele is correlated with higher levels of total cholesterol, LDL, apoB, and apoC [49]. Moreover, the study of men with metabolic syndrome showed that the frequency of the rs1800206 G allele was higher in subjects having simultaneously abdominal obesity, hypertriglyceridemia, and low HDL levels. The same study demonstrated that carriers of the G allele were characterized by higher plasma apoB and triglyceride (TG) levels and the presence of G allele was associated with components of metabolic syndrome [50]. In another relative large-scale study of middle-aged whites, the G allele was also correlated with an increase in fasting levels of serum lipids [51]. In a controlled dietary intervention trial, in which saturated fat was replaced with either monounsaturated fat or carbohydrate in isoenergetic diets, the effects of *PPARA* Leu162Val genotypes in the determination of plasma lipid concentrations were assessed. The results of this study revealed that Leu162Val variants influence plasma LDL cholesterol concentration, especially being a determinant of small dense LDL (sdLDL) [52]. It has been confirmed in several independent studies showing that rs1800206 CC homozygotes are characterized by a larger LDL particle with reduced density and by an increased general proportion of large LDL particles in the total cholesterol pool [53,54]. Such larger and more buoyant LDL particles are less prone to oxidation processes which create protecting conditions in case of atherosclerosis progression, while small dense LDLs are considered as risk factors of atherosclerosis and coronary artery disease [55,56]. Studies in patients demonstrated that fibrate ligands of PPARα can reduce production of sdLDL, so in carriers of the less active PPARα Val form (that is produced in rs1800206 G allele carriers), activation by dietary ligands could result in a shift to a higher proportion of sdLDL [47,57].

All the aforementioned studies led us to the suggestion that the *PPARA* rs1800206 G allele (producing PPARα protein with Val amino acid in the 162 position) may be associated with developing in its carriers the adverse effects in the context of lipid metabolism. Our results, at least in part, confirm this hypothesis, because the unfavorable post-training effects expressed by an increase of the HDL level was pointed out most firmly in GC heterozygotes, while in CC homozygotes these disadvantageous changes were significantly restricted.

We are aware that our study has some limitations. The first issue of almost every genetic association study has a proper number of participants in the study group. In our case, this could also be a problem, and we see the need for replicating our results in another, preferably larger, population. Especially, *PPARA* I7 rs4253778 CC genotype was rare (*n* = 6) and might cause statistical bias. On the other hand, this rs4253778 CC genotype has been rare in previous studies on the Caucasian population, where it was shown to influence physical condition level [58,59]. The second question is whether the analyzed *PPARA* polymorphisms are true causative factors or perhaps only in linkage disequilibrium with variants directly engaged in developing a specific trait. This problem has been brought up in many studies, and in most cases, the conclusion is that the variation within the *PPARA* gene does not influence any physiological traits alone. Thus, it should be underlined that *PPARA* diversity probably accounts for only a small portion of phenotypic variability, due to the polygenic character of the traits connected with body mass and biochemical parameters measured in our experiment, implying that multiple gene-environment interactions may contribute to the observed differential effects.

## 5. Conclusions

The results obtained in the current study support our initial hypothesis and suggest that *PPARA* intron 7 rs4253778, as well as Leu162Val rs1800206 variants, play a role in differentiating the beneficial effects of physical activity between the specific genotype carriers. We have demonstrated that harboring a specific *PPARA* intron 7 rs4253778 as well as Leu162Val rs1800206 genotypes may be associated with different post-training changes of measured biochemical parameters. We have observed the surprising effect of a post-training increase of plasma glucose concentration as well as the highest rise of LDL levels in rs4253778 CC participants, which led us to the suggestion that rs4253778 C allele may affect the lipid profile and glucose levels. On the other hand, we have also indicated that some individuals may benefit from being an rs1800206 CC homozygote because in such participants the unfavorable training effects were significantly restricted.

The information obtained in this study can be used as an additional source of precise information about a person undertaking physical effort, determining at the molecular level its inherent metabolic characteristics. Potentially, such information may help design individualized forms of training and more effective optimization and control of the obtained post-training or other treatment effects. Due to the importance of the polymorphic forms analyzed in *PPARA* gene in the etiology of many human diseases, they can be used as a molecular tool of pro-health prophylaxis, helpful in estimating the risk of disorders, such as obesity.

## Figures and Tables

**Figure 1 jcm-08-01043-f001:**
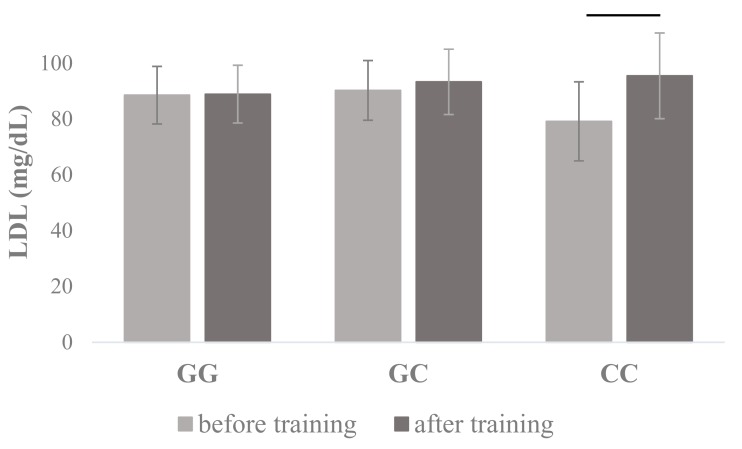
Changes in plasma low-density lipoproteins (LDL) concentrations measured before and after the completion of the 12-week training program in carriers of different *PPARA* I7 rs4253778 genotypes. The values are mean ± SD.

**Figure 2 jcm-08-01043-f002:**
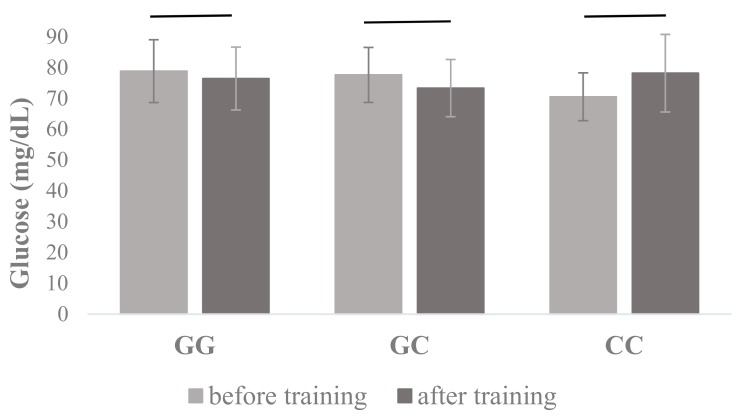
Changes in plasma glucose concentrations measured before and after the completion of the 12-week training program in carriers of different *PPARA* I7 rs4253778 genotypes. The values are mean ± SD.

**Figure 3 jcm-08-01043-f003:**
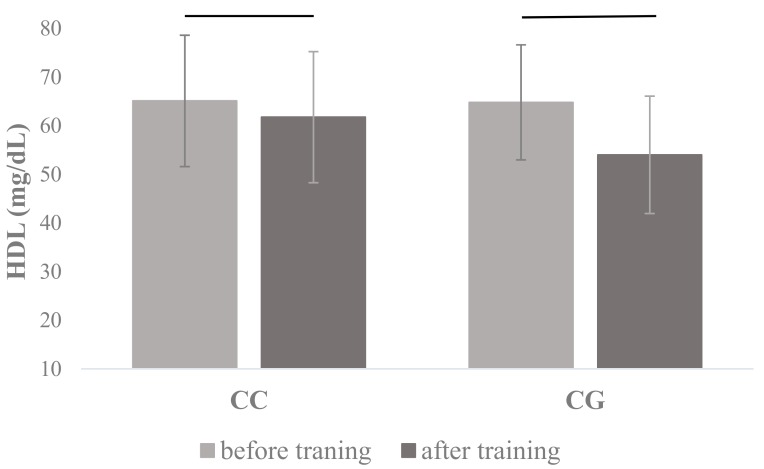
Changes in plasma high-density lipoproteins (HDL) concentrations measured before and after the completion of the 12-week training program in carriers of different *PPARA* Leu162Val rs1800206 genotypes. The values are mean ± SD.

**Table 1 jcm-08-01043-t001:** The *PPARA* I7 rs4253778 genotypes and response to training.

Variable	GG (*n =* 109)		GC (*n =* 53)		CC (*n =* 6)		*p* Values for Genotypes		*p* Values for Allele × Training Interaction	
	Before Training	After Training	Before Training	After Training	Before Training	After Training	Genotype	Training	GG vs. GC+CC	GG+GC vs. CC
Body mass (kg)	59.82 ± 7.45	59.12 ± 7.31	62.35 ± 7.93	61.51 ± 7.86	58.93 ± 6.92	58.27 ± 7.35	0.130	0.859	0.220	0.544
BMI (kg/m^2^)	21.36 ± 2.36	21.14 ± 2.30	22.06 ± 2.51	21.82 ± 2.48	21.47 ± 2.85	21.25 ± 2.87	0.226	0.984	0.964	0.510
BMR (kJ)	6027.11 ± 322.75	5988.31 ± 301.69	6117.87 ± 333.88	6087.24 ± 337.16	6035.00 ± 309.97	5970.17 ± 322.14	0.194	0.855	0.889	0.602
Tissue impedance (Ohm)	556.21 ± 64.49	542.09 ± 62.98	536.09 ± 62.33	524.34 ± 62.26	557.17 ± 50.41	543.50 ± 42.88	0.257	0.382	0.643	0.578
FM (kg)	14.42 ± 4.98	13.54 ± 4.97	15.67 ± 5.21	14.59 ± 5.35	14.32 ± 5.15	13.10 ± 5.53	0.565	0.647	0.955	0.979
FFM (kg)	45.38 ± 3.10	45.79 ± 3.06	46.45 ± 3.34	46.91 ± 3.58	44.62 ± 2.57	45.17 ± 2.38	0.084	0.936	0.874	0.491
TBW (kg)	33.32 ± 2.46	33.56 ± 2.28	33.84 ± 2.88	34.32 ± 2.69	32.67 ± 1.88	33.37 ± 1.51	0.229	0.481	0.813	0.429
Total cholesterol (mg/dL)	169.11 ± 22.86	167.16 ± 24.21	172.30 ± 29.33	170.53 ± 33.83	165.00 ± 10.18	172.67 ± 10.65	0.671	0.549	0.983	0.076
TGL (mg/dL)	81.90 ± 35.08	85.27 ± 36.75	76.15 ± 24.53	81.55 ± 32.75	85.50 ± 38.52	73.50 ± 15.54	0.825	0.475	0.679	0.414
HDL (mg/dL)	64.09 ± 12.29	61.21 ± 12.65	66.75 ± 15.18	60.86 ± 15.46	68.80 ± 14.95	62.28 ± 12.26	0.059	0.211	0.085	0.481
LDL (mg/dL)	88.54 ± 20.31	88.91 ± 20.34	90.26 ± 25.69	93.32 ± 29.70	79.17 ± 14.16	95.48 ± 15.35	0.648	**0.025**	0.171	**0.033**
Glucose (mg/dL)	78.83 ± 10.20	76.44 ± 10.21	77.62 ± 8.92	73.34 ± 9.28	70.50 ± 7.76	78.17 ± 12.58	0.298	**0.020**	0.101	**0.011**

The values are mean ± SD; *p* values (analyzed by two-way mixed ANOVA test) for main effects (genotype and training) genotype × training interaction; bold *p* values-statistically significant differences (*p* < 0.05); BMI–body mass index; BMR–basal metabolic rate; FM–fat mass; FFM–fat free mass; TBW–total body water; TGL–triglycerides; HDL–high-density lipoproteins; LDL–low-density lipoproteins.

**Table 2 jcm-08-01043-t002:** The *PPARA* Leu162Val rs18000206 genotypes and response to training.

Variable	CC (*n* = 154)		CG (*n* = 14)		*p* Values for Main Effects	
	Before Training	After Training	Before Training	After Training	Genotype	Training
Body mass (kg)	60.57 ± 7.69	59.90 ± 7.58	60.77 ± 7.37	59.21 ± 7.13	0.063	0.052
BMI (kg/m^2^)	21.55 ± 2.46	21.34 ± 2.39	22.03 ± 2.16	21.58 ± 2.39	0.0.64	0.099
BMR (kJ)	6055.80 ± 328.86	6021.35 ± 317.31	6058.43 ± 304.72	5991.64 ± 304.76	0.064	0.303
Tissue impedance (Ohm)	548.23 ± 62.99	535.32 ± 62.33	568.21 ± 71.46	549.93 ± 64.19	0.233	0.582
FM (kg)	14.73 ± 5.09	13.85 ± 5.10	15.63 ± 4.89	13.98 ± 5.34	0.098	0.163
FFM (kg)	45.77 ± 3.19	46.20 ± 3.25	44.79 ± 3.20	45.24 ± 3.15	0.716	0.090
TBW (kg)	33.52 ± 2.60	33.86 ± 2.41	32.81 ± 2.36	33.13 ± 2.32	0.096	0.950
Total cholesterol (mg/dL)	170.99 ± 24.67	169.36 ± 27.58	158.78 ± 23.55	158.07 ± 21.24	0.690	0.876
TGL (mg/dL)	80.86 ± 32.29	85.12 ± 35.55	73.14 ± 31.12	67.71 ± 22.64	0.900	0.294
HDL (mg/dL)	65.12 ± 13.51	61.78 ± 13.48	64.82 ± 11.83	54.03 ± 12.08	**0.001**	**0.013**
LDL (mg/dL)	89.60 ± 21.69	90.55 ± 23.39	79.43 ± 23.59	90.43 ± 26.16	0.259	0.587
Glucose (mg/dL)	78.41 ± 9.74	75.32 ± 10.02	75.36 ± 10.59	77.71 ± 10.67	0.795	0.053

The values are mean ± SD; *p* values (analyzed by two-way mixed ANOVA test) for genotype × training interaction; BMI–body mass index; BMR–basal metabolic rate; FM–fat mass; FFM–fat free mass; TBW–total body water; TGL–triglycerides; HDL–high-density lipoproteins; LDL–low-density lipoproteins.

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
