# Peer review of "The Polymorphisms of the Peroxisome-Proliferator Activated Receptors’ Alfa Gene Modify the Aerobic Training Induced Changes of Cholesterol and Glucose"

_jcm, 2019, doi:10.3390/jcm8071043_

Round 1
Reviewer 1 Report
The principal aim of this study was to check if post-training changes of body mass measurements as well as chosen biochemical parameters are modulated by the PPARA genotypes. The results point that PPARA intron 7 rs4253778 CC genotype modulate training response by increasing LDL and glucose concentration, while PPARA Leu162Val rs1800206 CG genotype polymorphism interacts in decrease in HDL concentration. The authors concluded Carriers of PPARA intron 7 rs4253778 CC genotype and Leu162Val rs1800206 CG genotype might have potential negative training-induced cholesterol and glucose changes after aerobic exercise. The study is well-designed and well-conducted, and the results are interesting and substantial for the community. However, authors should provide some information to refine their article.
Comments:
Introduction: Lines 109-113: Even if the introduction is well written, I don’t understand why the authors approach their statistical treatments and results in the introduction section. I suggest that the authors remove this part of the introduction.
Experimental Section:
- Line 126: The authors state that of the 201 participants, none are engaged in regular physical activity during the last 6 months. I would like to know how they measured the level of physical activity over the last 6 months.
- Lines 148-153: In the experimental protocol, the intensity of the program was gradually increased based on the maximum heart rate. I would like to know the formula the authors used to calculate the heart rate. In addition, I would like to know why the authors did not use the reserve heart rate instead of the maximum heart rate.
- Line 168: according to my knowledge, the unity of body mass index is kg/m² or Kg.m-2. I don't understand why the authors wrote this: (m2)-1). To avoid any confusion, I suggest that the authors use the international unity system.
Line 202: As I understand it, the authors performed anthropometric and biochemical measurements in 182 participants. In contrast, for the genetic analysis, it was carried out in 168 participants. How the authors explain this experimental loss (reduction of participants number?)
Results:
- Even if the results are interesting, I wonder if the unequal size of the groups (confers Table 1 and 2) does not constitute an important bias for the statistical analysis carried out. for example, in Table 1 it can be noted that the CC group is composed of 6 participants if I am not mistaken. The authors should therefore emphasize this point in the limits of the study.
Discussion:
- Lines 295-298: The authors cited the reference [40] as a meta-analysis, but when the reference [40] is checked in the bibliographic section, this is not the case at all. This reference [40] is an original article [Differences in Transcriptional Activation by the Two Allelic (L162V Polymorphic) Variants of PPARα after Omega-3 Fatty Acids Treatment]. I suggest that authors check the bibliographic references carefully.
Conclusion: I suggest that the authors write a very small paragraph on perspectives in this section.
Author Response
Reviewer 1
The principal aim of this study was to check if post-training changes of body mass measurements as well as chosen biochemical parameters are modulated by the PPARA genotypes. The results point that PPARA intron 7 rs4253778 CC genotype modulate training response by increasing LDL and glucose concentration, while PPARA Leu162Val rs1800206 CG genotype polymorphism interacts in decrease in HDL concentration. The authors concluded Carriers of PPARA intron 7 rs4253778 CC genotype and Leu162Val rs1800206 CG genotype might have potential negative training-induced cholesterol and glucose changes after aerobic exercise. The study is well-designed and well-conducted, and the results are interesting and substantial for the community. However, authors should provide some information to refine their article.
Answer: Thank you for this detailedfeedback we have improved our article according your suggestion.
Comments:
Introduction: Lines 109-113: Even if the introduction is well written, I don’t understand why the authors approach their statistical treatments and results in the introduction section. I suggest that the authors remove this part of the introduction.
Answer: We agree, this was an accident and we removed this to the results section.
Experimental Section:
- Line 126: The authors state that of the 201 participants, none are engaged in regular physical activity during the last 6 months. I would like to know how they measured the level of physical activity over the last 6 months.
Answer: The level of physical activity over the last 6 months has been estimated in every participant according to subjective methods via Global Physical Activity Questionnaire (GPAQ) as well as individual recording of subject’s own activity such as direct observation and activity diaries. We reference this kind of estimation (Hill 2014).
- Lines 148-153: In the experimental protocol, the intensity of the program was gradually increased based on the maximum heart rate. I would like to know the formula the authors used to calculate the heart rate. In addition, I would like to know why the authors did not use the reserve heart rate instead of the maximum heart rate.
Answer: Thank you, this information was really missing and now we described the measurement process for maximum heart rate. Maximum heart rate (HRmax) was measured directly (not calculated) in every subject since a continuous graded exercise test on an electronically braked cycle ergometer has been performed during last week screening session prior to the training program to determine their aerobic capacity (VO2max).
After the reviewers’ comments we have carefully checked the details and have found mistakes in the description part of the exercise intensity. In fact, we have used the reserve heart rate (HRR) to calculate and monitor the intensity of the training program, which was set up for participants comfort in Polar system according to Karvonen formula. Thus, all mistakes have been corrected in the manuscript.
- Line 168: according to my knowledge, the unity of body mass index is kg/m² or Kg.m-2. I don't understand why the authors wrote this: (m2)-1). To avoid any confusion, I suggest that the authors use the international unity system.
Answer: Done, we corrected the units and BMI calculation.
Line 202: As I understand it, the authors performed anthropometric and biochemical measurements in 182 participants. In contrast, for the genetic analysis, it was carried out in 168 participants. How the authors explain this experimental loss (reduction of participants number?)
Answer:
We agree, this was confusing in first version. Now we described the drop of participant at beginning of “participant description paragraph and put only final number of participants (168) into abstract and other parts of the text.
From 201 participants that have completed the training program, we have obtained the full set of selected body mass and body composition variables before and after the completion of a 12-week training period only for 182 participants. From these 182 participants the genetic material has been isolated. The DNA isolates have been verified for DNA amount, integrity and purity – after the verification only 168 samples were used for genotyping. Now we made this very clear in participant section and further we are using just n = 168.
Results:
- Even if the results are interesting, I wonder if the unequal size of the groups (confers Table 1 and 2) does not constitute an important bias for the statistical analysis carried out. for example, in Table 1 it can be noted that the CC group is composed of 6 participants if I am not mistaken. The authors should therefore emphasize this point in the limits of the study.
Answer: We agree with this point and we added this fact in study limitations. We have added also the references, where this CC genotype is often rare and influence the condition level. Moreover, according to second reviewer comments we amend it the statistical report for unequal effect sizes. Now we also reported partial eta square and degrees of freedom.
Discussion:
Answer:
- Lines 295-298: The authors cited the reference [40] as a meta-analysis, but when the reference [40] is checked in the bibliographic section, this is not the case at all. This reference [40] is an original article [Differences in Transcriptional Activation by the Two Allelic (L162V Polymorphic) Variants of PPARα after Omega-3 Fatty Acids Treatment]. I suggest that authors check the bibliographic references carefully.
Answer: We have carefully checked the bibliographic references and have discovered that there was a mistake from the position [30] in the reference list. Thus, the in-text citations have been corrected as well as the whole reference list has been reorganized.
Conclusion: I suggest that the authors write a very small paragraph on perspectives in this section.
Answer: This paragraph on perspectives has been added to the conclusions.
Reviewer 2 Report
The manuscript is well written and clear. The language is acceptable, however there are few typos and grammar mistakes requiring correction. However there are some issues that need some attention.
I would have two more important concerns:
1. regarding the methodology and results: taking into account relatively small number of infrequent SNPs homozygotes is it possible that significant differences in LDL, HDL and glucose results may be affected by the diet, which was not very strictly determined or other random factors? This is especially seen for diverse glucose levels.
2. Please provide info about the data values provided in the text and graphs: are these means +/- SD or SEM?
Minor remarks:
In methods (lines 149-153) authors state that the training program assumed certain HR levels at each step, but there is no information if HR was measured at all. Was it measured?
Methods, statistical analysis: was the sphericity assumption tested?
Line 276: I was not able to find the overall effect of training on the BW, BMI, TBW, FM, impedance in the results (and tables do not present any significant changes), so it should not be stated in the discussion in this way.
Author Response
The manuscript is well written and clear. The language is acceptable, however there are few typos and grammar mistakes requiring correction. However, there are some issues that need some attention.
I would have two more important concerns:
1. regarding the methodology and results: taking into account relatively small number of infrequent SNPs homozygotes is it possible that significant differences in LDL, HDL and glucose results may be affected by the diet, which was not very strictly determined or other random factors? This is especially seen for diverse glucose levels.
Answer: The diet has been carefully set up during the study and 2 months before the study (line 124). We have added the time period to participants get familiarized with dietary intake.
2. Please provide info about the data values provided in the text and graphs: are these means +/- SD or SEM?
Thank you for this detailed feedback we have improved our article according your suggestion.
Answer: The data values provided in the text and graphs are mean ± SD. The appropriate changes have been made to the text.
Minor remarks:
In methods (lines 149-153) authors state that the training program assumed certain HR levels at each step, but there is no information if HR was measured at all. Was it measured?
Answer: The HR at each step of the training program was measured in every subject with using HR personal monitoring devices, which is now described in methods.
Methods, statistical analysis: was the sphericity assumption tested?
Answer:
To the best of our knowledge, the sphericity is appropriate to be taken when looking at multiple (more than 2) measurements (especially more than 2 repeated measures). We used only one within between subject factor and one between subject factor. That’s why the sphericity assumption was not tested in our first analyses, because there are only two sets of measurements for every variable (before and after the training). However, we now stated that sphericity has not been interrupted by Mauchly's test.
In relation to this point we decided to report results in ANOVA for repeated measures and add more statistical values, such as effect size by partial eta square. This resulted in changes in results reporting section and correction of p values, however no changes in results were performed since just different ANOVA type did not change main effects..
Line 276: I was not able to find the overall effect of training on the BW, BMI, TBW, FM, impedance in the results (and tables do not present any significant changes), so it should not be stated in the discussion in this way.
Answer: We agree, there were just insignificant effects sizes observed, which should not be reported like this even in discussion. Therefore, the paragraph regarding an overall effect of training on the measured variables has been removed from the discussion.
Round 2
Reviewer 1 Report
Good job. Thanks for your answers.